# Serological survey in a university community after the fourth wave of COVID-19 in Senegal

**Fatou THIAM**[1]*, **Abou Abdallah Malick DIOUARA**[1], **Clemence Stephanie Chloe Anoumba NDIAYE**[1], **Ibrahima DIOUF**[2], **Khadim KEBE**[1], **Assane SENGHOR**[1], **Djibaba DJOUMOI**[1], **Mame Ndew MBAYE**[1], **Idy DIOP**[3], **Sarbanding SANE**[1], **Seynabou COUNDOUL**[1], **Sophie Deli TENE**[1], **Mamadou DIOP**[1], **Abdou Lahat DIENG**[2], **Mamadou NDIAYE**[4], **Saidou Moustapha SALL**[2], **Massamba DIOUF**[5], **Cheikh Momar NGUER**[1]

1 Groupe de Recherche Biotechnologies Appliquées & Bioprocédés Environnementaux, École Supérieure Polytechnique, Université Cheikh Anta Diop de Dakar, Dakar, Senegal, 2 Laboratoire Physique de l'Atmosphère et de l'Océan-Siméon Fongang, École Superieure Polytechnique, Université Cheikh Anta Diop de Dakar, Dakar, Senegal, 3 Laboratoire d'Imagerie Médicale et de Bio-Informatique, École Superieure Polytechnique, Université Cheikh Anta Diop de Dakar, Dakar, Senegal, 4 Laboratoire Mathématiques Appliquées et Informatique, Faculté des Sciences et Techniques, Université Cheikh Anta Diop de Dakar, Dakar, Senegal, 5 Laboratoire Sante Publique, Institut d'odontologie et de Stomatologie, Faculte de Medecine, de Pharmacie et d'Odonthologie, Université Cheikh Anta Diop de Dakar, Dakar, Senegal

* fatou54.thiam@ucad.edu.sn

**Data Availability Statement:** All data are in the manuscript and/or Supporting information files.

## Abstract

A cross-sectional survey was conducted at Polytechnic High School (PHS) to assess the spread of COVID-19 infection among students and staff. A random cluster sampling was conducted between May 19 and August 18, 2022, after the fourth wave of COVID-19 in Senegal. IgM and IgG SARS-CoV-2 antibodies were screened using WANTAI SARS-CoV-2 ELISA assays. Seroprevalence and descriptive statistics were calculated, and associations between seropositivity and different factors were determined using logistic regression. A total of 637 participants were recruited and the median age was 21 years [18–63]. 62.0% of the participants were female, and 36.89% were male, with a male-to-female ratio = 0.59. The overall IgG and IgM seroprevalence were 92% and 6.91% respectively. Among those who tested positive for IgM, 6.75% were also positive for IgG, and 0.15% were negative for IgG. Interestingly, 6.90% of participants tested negative for both IgM and IgG. We found a higher IgM seroprevalence in men than women (9.4% vs. 5.6%) and a lower IgM seroprevalence in (18–25) age group compared to (55–65) years. We revealed a significant difference according to IgG seroprevalence among participants who declared fatigue symptoms [92.06% (95% CI: 89.96–94.16)] compared to those who did not [80.39% (95% CI: 77.31–83.47)], $p = 0.0027$. IgM seropositivity was found to be associated with Body Mass Index (BMI) categories (O.R. 0.238, p = 0.043), ethnic group (O.R. 0.723, $p = 0.046$), and marital status (O.R. 2.399, $p = 0.021$). Additionally, IgG seropositivity was linked to vaccination status (O.R. 4.741, $p < 0.001$). Our study found that most students and staff at PHS were exposed to SARS-CoV-2, confirming the virus's circulation at the time of the survey. We also identified differences in individual susceptibility that need further clarification. Our results highlight the importance of seroepidemiological surveys to

**Funding:** This study was financially supported by Cheikh Anta Diop University of Dakar's (UCAD) [https://www.ucad.sn] Ecole Supérieure Polytechnique (ESP) [https://esp.sn] in the form of a Research Impulse Fund award received by FT. This study was also financially supported by UCAD ESP in the form of salaries for AAMD, AS, MNM, ID, MD, ALD, SMS and CMN. The specific roles of these authors are articulated in the 'author contributions' section. No additional external funding was received for this study.

**Competing interests:** The authors have declared that no competing interests exist.

assess the true impact of the COVID-19 pandemic in a community and to monitor variations in antibody response.

## Introduction

The Coronavirus Disease 2019 (COVID-19) was first identified in December 2019 in Wuhan, China. It is an infectious disease caused by the Severe Acute Respiratory Syndrome Coronavirus 2 (SARS-CoV-2) [1, 2]. On January 30 2020, the World Health Organization (WHO) declared COVID-19 a public health emergency of global concern, and on March 11, 2020, it announced the COVID-19 epidemic as a pandemic [3]. By November 12, 2023, more than 697 million people had been infected worldwide, with over 6.9 million deaths [4]. The COVID-19 outbreak had substantial economic, social, and health repercussions and damaging educational consequences [5–7]. In April 2020, the World Bank estimated that higher education institutions had been closed in 175 countries and that studies had been interrupted or significantly disrupted due to COVID-19, affecting more than 220 million students [8]. The pandemic forced academic communities to adopt online platforms for the continuity of teaching and learning activities, negatively impacting learning outcomes, particularly in developing countries where the lack of network infrastructures, computers, and internet access is challenging distance learning in developing countries [9, 10].

In Senegal, a West African country, the first case of SARS-CoV-2 was identified on March 2, 2020. Since then, the number of cases has risen considerably, and the country currently has more than 89,022 confirmed cases of COVID-19 and 1,971 deaths [11]. The onset of the pandemic led the government authorities to close higher education establishments from March 16 to August 31 2020. The establishments reopened with the implementation of sanitary protocols that emphasized the reinforcement of barrier measures. These measures included wearing masks, washing hands with soapy water or using hand sanitiser, maintaining social distancing, and taking body temperature at campus access points [12]. However, these measures have often been insufficient to stem the spread of the virus. Clusters quickly appeared in some establishments, including the Cheikh Anta Diop (CAD) University, located in Dakar, the capital of Senegal.

The failure of strategies to control the virus spread could be related to various factors, including the CAD University ecosystem, which accommodates over 80,000 students from diverse origins. The consequence is overcrowding in student residences that caused the virus spread; (ii) the university environment is mainly populated by young people, who do not seem to be fully aware of the risks associated with the COVID-19 health crisis [13]; (iii) the promiscuity in the Halls of residence, making it difficult to comply with social preventive measures; (iv) Lastly, lack of vaccine deployment and reluctance was noted among young people, which also contributes to low vaccination coverage in universities [14].

Faced with all these difficulties, it was essential to adopt cyclical measures to manage the COVID-19 pandemic, characterised by the virus spreading in waves [15, 16]. These measures could involve introducing survey studies based on diagnostic testing and mass screening protocols to diagnose and follow up with people exposed to or infected with SARS-CoV-2. In practice, detecting viral RNA by RT-PCR is the reference method for confirming the diagnosis of SARS-CoV-2 infection [17, 18]. However, access to diagnostic tests still needs to be improved in poor countries [19]. The other way to estimate the true extent of the epidemic is to conduct Seroprevalence surveys [20, 21]. Seroepidemiological studies to detect the presence

of anti-SARS-CoV-2 antibodies are a valuable tool for assessing the timing of the epidemic. It can help to confirm the presence of a recent infection when PCR is limited. Some antibodies, such as IgG, can even be detected years after exposure [22]. Their detection can be used for tracking the spread of infection and defining herd immunity barrier and individual immunization levels in the ongoing COVID-19 pandemic [23]. By carrying out this analysis on a representative population, it is possible to estimate what proportion of the population has already been exposed to the new coronavirus [24]. This information helps identify the epidemic phases and can help the authorities make decisions and even anticipate appropriate measures to contain the pandemic's spread [25]. Sero-epidemiological studies have been conducted in academic institutions such as universities in many countries [26–29]. However, more studies are highly relevant because university communities (faculty, staff and students) could be among the most exposed to SARS-CoV-2. In Senegal, no seroprevalence studies have been conducted in cohorts of educational institutions.

We conducted an on-site screening project at Polytechnic High School (PHS), a public institution with an inter-African focus at CAD University, from May to August 2022, following the fourth wave of the Coronavirus disease in Senegal. Our goal was to assess the true extent of previous and recent COVID-19 exposure among students and staff and investigate the risk factors associated with SARS-CoV-2 IgM and IgG seropositivity.

## Materials and methods

### Study design and population

The SARSESP ("Etude de Seroprevalence du SARS-CoV-2 au sein de l'École Su-périeure Polytechnique", in the Cheikh anta DIOP University (CAD University)) project is an on-site university population-based cross-sectional study. The sampling occurred at the PHS, an establishment of CAD University. Students, Professors, and Technicians, Administrative and Service (TAS) officers at SPS were invited by e-mail to enrol in the study. Participants were volunteers who registered online between 19 May and 18 August 2022. A questionnaire was administered to each participant after consent, blood samples were taken for SARS-CoV-2 antibody detection.

### Sample size calculation

We used stratified random sampling: the first stratum concerned professors, the second TAS officers and the third students. Systematic random sampling was used within each stratum to determine the required subjects number. Using the lists provided by the student affairs and human resources departments, we calculated the sample size based on the hypothesis of an expected seroprevalence of 45%, with a precision of 5%, a design effect of 1.96, and a nonresponse rate of 65%. We determined that >450 participants needed to be recruited. To increase the accuracy of the results, this size was multiplied by 2, giving a sample size of 898, and rounded up to 1000 to account for any lost records. The survey step for selecting statistical units was 6133/1000, i.e. a step = 6. In this study, an allocation proportional to the size of each stratum was used. Thus, we considered a proportion of 83.54% for students, 9.97% for professors and 6.47% for TAS officers. Then, we obtained 65 TAS officers, 100 professors and 835 students. The inclusion criteria were age over eighteen (18) years, informed consent signed, and questionnaire completion. Then, the non-inclusion criteria were under 18 years of age, non-consent and contraindications to venous blood sampling (anaemia, Infection or hematoma at a prospective venipuncture site, etc.).

## Questionnaire

We shared an interviewer-administered questionnaire with participants on an electronic tablet (https://enquete.ucad.sn/index.php/598685?lang=fr). The questionnaire covered points regarding different factors to assess for relationships between IgM and IgG seropositivity and these factors. Questions relating to sociodemographic characteristics were collected, such as age, sex, occupation, education level, nationality, ethnic group and accommodation type. We also collected alcohol and tobacco intake, SARS-CoV-2 vaccination status, and preventive measures related to SARS–CoV–2 practices. We provided all recruitment participants with face masks and hand sanitisers and encouraged them to practice physical and social distancing. The questionnaire on COVID-19 was posted online by the IT and Information Systems Department of CAD University.

## Blood collection and SARS-CoV-2 antibodies detection

After each participant signed the written consent form, a 10 Ml whole blood sample was collected into a dry vacutainer tube by standard venipuncture technique. Blood samples were centrifuged at 2,500 rpm for 10 minutes. Then, the plasma was collected and stored in cryotubes at -80˚C at the GRBA-BE laboratory biobank in PHS until the tests were carried out. Seropositivity to anti-SARS-CoV-2 antibodies was used as a biomarker of exposure to the SARS-CoV-2 virus. IgM antibodies emerge early during immune responses (usually from days 5–7 after symptoms appear, but sometimes later and decrease at days 15–22), while IgG antibodies typically appear later (detectable from day 11 post symptom, reaching a maximum 3–4 weeks after) and exist in human bodies for months [23, 30, 31].

Serological tests were performed by qualitative ELISA for IgM and quantitative ELISA for IgG following the instructions for the WANTAI SARS-CoV-2 IgM ELISA (Beijing Wantai Biological Pharmacy Enterprise, Beijing, China; Ref. WS-1196 and WS-1396) recommended by the WHO for seroepidemiological studies, which detects total antibodies (including IgM and IgG) binding the SARS-CoV-2 spike protein receptor binding domain (S1/RBD) [32]. Serum samples were analysed in duplicate according to the supplier's recommendations.

## Statistical analyses

Statistical analyses were performed using Rstudio (version R 4.2.1) and GraphPad Prism (version 10.1.1) softwares. Continuous variables were described as mean (standard deviation) or median (interquartile range). Normally distributed variables were compared with a t-test, and nonparametric data were compared with the Mann-Whitney test. Categorical variables were presented as per cent, and Fisher exact tests or chi-squared tests were used for proportional assessments. For all statistical tests, we accepted a two-sided level of significance was set at $p \leq 0.05$.

SARS-CoV-2 seroprevalence was defined as the ratio of the number of people who developed anti-SARS-CoV-2 antibodies to the general population. Confidence intervals (95% CI) for seroprevalence were estimated using the Clopper-Pearson method. Using logistic regressions, we sought to understand how the IgM and IgG seropositivity are influenced by various variables such as age, gender, occupation, education level, nationality, type of accommodation, etc. The analyses were carried out after carefully handling the database and processing missing data.

## Ethics statement

This research complies with ethical recommendations. The project protocol has been validated by the National Health Research Ethics Committee (CNERS) of the Ministry of Health and

Social Action. The research protocol was drawn up following Senegalese laws and regulations governing the confidentiality of personal data. The study was approved by the Senegalese National Ethics Committee for Research in Health (Reference number N˚000043/MSAS/CNERS/SP, 28 February 2022).

## Results

### Baseline characteristics of participants

From the 1000 students, professors and TAS officers of PHS who were invited to participate in the study between 19 May 2022 and 18 August 2022, 637 (63.7%) participants were finally included (Fig 1). The most represented group was students (88.38%), followed by TAS officers (8.16%), and professors (3.45%). The mean age was 23.20 years, ranging from 18 to 63 years. The population's median age was 21 years, with most participants aged 18 to 25 years, representing 85.7% of the population (Table 1). 62.0% of the participants were female, and 36.89% were male, with a male-to-female ratio = 0.59. Most participants were Senegalese (94.66%) and lived in family homes (54.6%). In addition, most ethnic groups in the PHS population were Wolof (31.24%), Fula (24.33%) and Serer (21.98%). The prevalences of active smoking and alcohol consumption were relatively low in the PHS community at the moment of the survey (1.9%) and (3.62%), respectively. Calculation of body mass index also revealed the highest prevalence of normal weight (37.68%), followed by underweight (10.8%) and overweight/obesity, 8.32%). The most common blood group is O+ (47.08%). At the time of data collection, unvaccinated *vs*. vaccinated participants were 57.46% and 35.01% respectively.

### Seroprevalence of IgM et IgG SARS-CoV-2 antibodies

Out of all participants, 6.91% (95% CI: 4.93–8.87) were seropositive for SARS-CoV-2 IgM antibody, and 92% (95% CI: 89.90–94.11) were seropositive for SARS-CoV-2 IgG (Table 2). Among those who tested positive for IgM, 6.75% were also positive for IgG, and 0.15% were negative for IgG. interestingly, 6.90% of participants tested negative for both IgM and IgG (S1 Fig). Finally, we found that 83.67% were IgM negative and IgG positive. Then, we also analysed IgM and IgG seroprevalence according to the different characteristics of the study population.

The IgM seroprevalence results showed significant age-dependence ($p < 0.001$) with higher seroprevalence in the (25–35) age group with 23.69% (95% CI: 20.39–27) compared to (18–25) age group with 5.5% (95% CI: 3.73–7.27) (Table 2). According to gender, IgM seroprevalence was higher in men, 9.4% (95% CI: 7.13–11.67) than in women, 5.6% (95% CI: 3.51–7.38), but no significant differences were found by sex ($p = 0.83$). There were no significant differences according to the other sociodemographic parameters regarding IgM antibodies seropositivity, such as occupation, level of education, ethnic group, accommodation type and size, but with some differences between sub-groups. For example, depending on the type of residence, we found a higher IgM seroprevalence among those living in halls of residence, 7.97% (95% CI: 5.87–10.07) compared to those living in family homes, 6.68% (95% CI: 4.74–8.62)], $p = 0.73$.

Regarding IgG seroprevalence, we found significant differences between those who drink alcohol (95.24%, 95% CI: 93.58–96.89), those who don't (90.80%, 95% CI: 88.55–93.04) and those who stopped drinking more than a year ago (66.66%, 95% CI: 64.35–68.07) with a $p = 0.038$ (Table 2). Additionally, a very significant difference was found based on vaccination status, with a higher seropositivity rate among vaccinated individuals (96.83%, 95% CI: 95.47–98.19) compared to unvaccinated individuals (86.58%, 95% CI: 83.93–89.22), $p < 0.0001$. No significant difference was found concerning IgG seroprevalence across different age groups. In terms of gender, IgG seroprevalence was lower in men (89.9%, 95% CI: 87.60–92.24) compared to women (90.6%, 95% CI: 88.33–92.87), with a p-value of 0.98. No significant

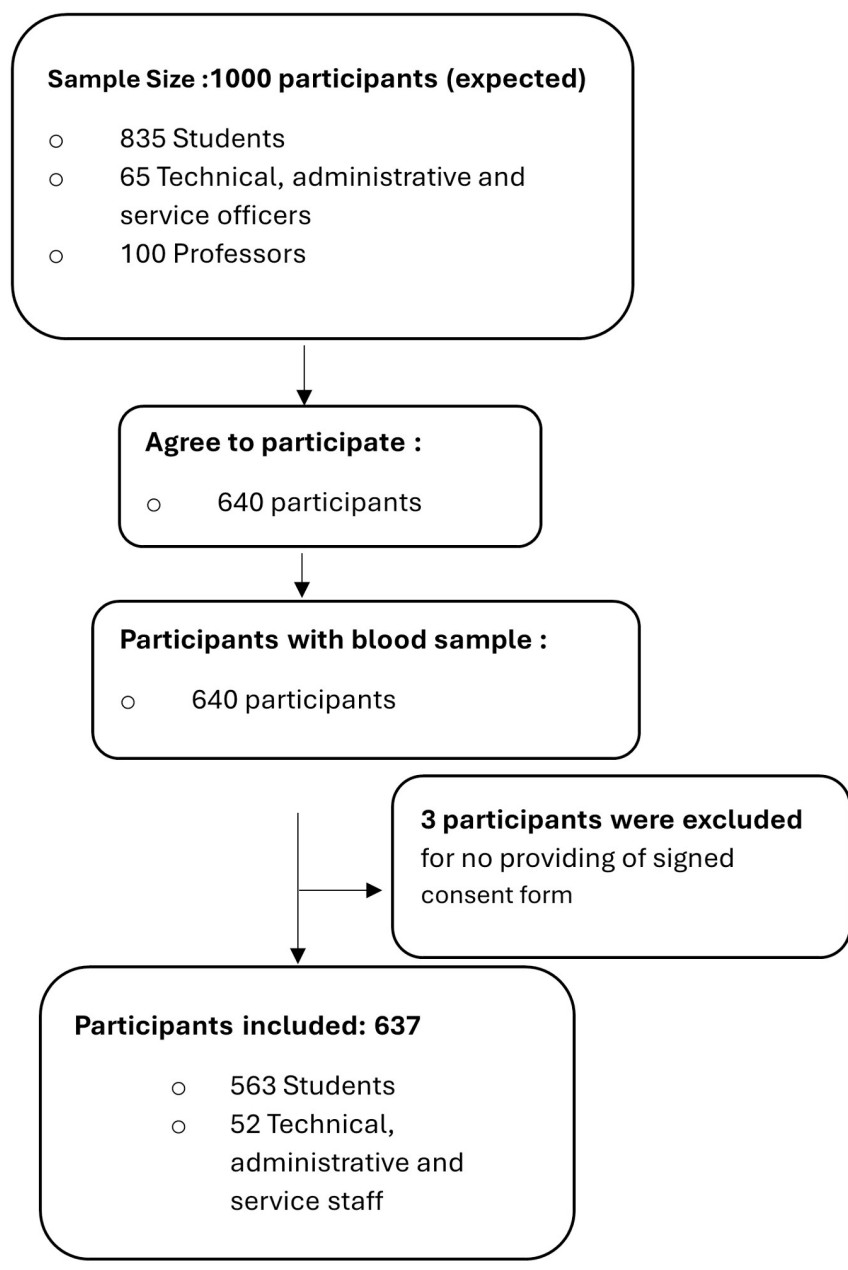

**Fig 1. Flowchart of participants' enrolment for the anti-SARS-CoV-2 antibody seroprevalence study in Superior Polytechnic School from May 19 to August 18, 2022.**

differences were found by sex (p = 0.83). Furthermore, while IgM and IgG seroprevalences were higher in professors and TAS officers than students, this difference was not statistically significant (p = 0.71).

## Compliance with preventive measures and antibodies' seroprevalence

Many respondents did not comply with the recommended preventive measures. Only 31.39% reported wearing masks often or always, and 18.21% maintained a physical distance of at least

**Table 1. Baseline characteristics of the study population.**

| Variable | Value, N (%) |
|---|---|
| **Overall** | 637 (100) |
| **Age (years)** | |
| Mean | 23.20 |
| Median | 21 |
| Range | 18–63 |
| **Age group (years)** | |
| [18–25] | 546 (85.71) |
| (25–35] | 38 (5.96) |
| (35–45] | 26 (4.08) |
| (45–55] | 12 (1.88) |
| (55–65] | 6 (0.94) |
| **Gender** | |
| Female | 395 (62) |
| Male | 235 (36.89) |
| **Occupation** | |
| Students | 563 (88.38) |
| Professors | 22 (3.45) |
| TASO | 52 (8.16) |
| **Level of education** | |
| Bachelor degree | 450 (70.64) |
| Master degree | 117 (18.36) |
| PhD | 20 (3.14) |
| others | 18 (2.82) |
| **Nationality** | |
| Senegalese | 603 (94.66) |
| Others † | 32 (5.03) |
| **Ethnic groups** | |
| Wolof | 153 (24.02) |
| Fula | 139 (21.82) |
| Serer | 79 (12.40) |
| Jola | 16 (2.50) |
| Malinke | 17 (2.67) |
| Soninke | 9 (1.41) |
| Mauri | 7 (1.1) |
| Others † | 18 (2.82) |
| **Accommodation type** | |
| Halls of residence | 138 (21.67) |
| Family home | 479 (75.19) |
| **Family members** | |
| **Family members** | |
| 1–2 | 37 (5.81) |
| 3–5 | 247 (38.77) |
| 6–8 | 188 (29.51) |
| 9 or plus | 74 (11.61) |
| **Matrimonial status** | |
| Single | 558 (87.60) |
| Married | 60 (9.42) |

(*Continued*)

**Table 1.** (Continued)

| Variable | Value, N (%) |
|---|---|
| Divorced | 2 (0.31) |
| **Smoker** | |
| Yes | 10 (1.88) |
| No | 538 (84.30) |
| Stopped > 1 year | 10 (1.41) |
| **Alcohol consumer** | |
| Yes | 21 (3.62) |
| No | 554 (86.81) |
| Stopped > 1 year | 3 (0.31) |
| **BMI Categorized** | |
| Underweight | 69 (10.83) |
| Normal weight | 240 (37.68) |
| Overweight /Obesity | 53 (8.32) |
| **Blood Group** | |
| A- | 9 (1.41) |
| A+ | 140 (21.98) |
| AB- | 1 (0.15) |
| AB+ | 25 (3.92) |
| B- | 5 (0.78) |
| B+ | 102 (16.01) |
| O- | 24 (3.77) |
| O+ | 300 (47.08) |
| **COVID-19 Vaccination** | |
| Yes | 221 (35.01) |
| No | 365 (57.46) |
| Prefer not to say | 51 (7.54) |

N number; BMI Body Mass Indice; TASO technicians, administrative and service officers

[†] For nationality and ethnic group, others correspond to others of African nationality or ethnic group with very small numbers (maximum 5).

2 meters over the last 15 days. However, most participants did comply with hand washing, with 75.35% reporting frequent hand washing (Fig 2A). Regarding social distancing, 71.58% reported participating in 0–2 social events, 70.8% used public transport 0–2 times a day, and 62.16% visited someone in the 15 days before the survey (Fig 2B). Antibody seroprevalence revealed significant IgM seropositivity related to the number of visits made to someone in the 15 days before the survey ($p$ = 0.008). However, there was no significant difference in IgG sero-positivity and compliance with barrier measures (S1 Table).

## Diagnosis and clinical symptoms related to COVID-19 and antibodies' seroprevalence

During the survey, a low rate of respondents, 8.95% declared to have tested positive by RT-PCR for COVID-19 infection since the beginning of the pandemic, while 82.89% declared to have tested negative. 8.16% did not disclose their test results (Fig 3A). 8.10% of respondents reported being diagnosed as COVID-19 positive by a healthcare professional, and 29.98% reported self-diagnosing as COVID-19 positive. A significant number of participants

**Table 2. Distribution of SARS-CoV-2 Ig M and G seropositive and seronegative individuals among the 637 participants enrolled.**

| Variable | IgM | | | IgG | | |
|---|---|---|---|---|---|---|
| | Seronegative N (%) | Seropositive N (%) | p-value | Seronegative N (%) | Seropositive N (%) | p-value |
| **Overall** | 593 (93.09) | 44 (6.91) | | 51 (8) | 576 (92) | |
| **Age (years)** | | | | | | |
| Mean | 23.06 | 25.16 | | 22.4 | 23.28 | |
| Median | 21 | 21 | | 21 | 21 | |
| Range | 18–63 | 18–59 | | 18–47 | 18–63 | |
| **Age group** | | | <**0.001*** | | | 0,98 |
| [18–25] | 516 (94.50) | 30 (5.5) | | 53 (9.7) | 493 (92.3) | |
| (25–35] | 29 (76.31) | 9 (23.69) | | 4 (10.53) | 34 (89.47) | |
| (35–45] | 24 (92.31) | 2 (7.69) | | 2 (7.69) | 24 (92.31) | |
| (45–55] | 12 (100) | 0 (0) | | 1 (8.33) | 11 (91.67) | |
| (55–65] | 4 (66.67) | 2 (33.34) | | 0 (0) | 6 (100) | |
| **Gender** | | | 0.16 | | | 0.89 |
| Female | 373 (94.43) | 22 (5.57) | | 37 (9.36) | 358 (90.63) | |
| Male | 213 (90.64) | 22 (9.36) | | 24 (10.21) | 211 (89.79) | |
| **Occupation** | | | 0.15 | | | 0.71 |
| Students | 528 (93.78) | 35 (6.22) | | 55 (9.77) | 508 (90.23) | |
| Professors | 19 (86.36) | 3 (13.64) | | 1 (9.61) | 21 (90.38) | |
| TASO | 46 (88.46) | 6 (11.54) | | 5 (4.54) | 47 (95.45) | |
| **Level of education** | | | 0.18 | | | 0.45 |
| Bachelor degree | 423 (94) | 27 (6) | | 46 (10.22) | 404 (89.78) | |
| Master degree | 105 (84.74) | 12 (10.26) | | 12 (10.25) | 105 (89.75) | |
| PhD | 17 (85) | 3 (15) | | 1 (5) | 19 (95) | |
| others | 16 (88.89) | 2 (11.11) | | 0 (0) | 18 (100) | |
| **Nationality** | | | 0.29 | | | 0.30 |
| Senegalese | 565 (93.39) | 40 (6.61) | | 56 (9.26) | 549 (90.74) | |
| Others † | 26 (86.67) | 4 (13.33) | | 5 (16.67) | 25 (83.33) | |
| **Ethnic groups** | | | 0.25 | | | 0.17 |
| Wolof | 145 (94.77) | 8 (5.23) | | 14 (9.15) | 139 (90.85) | |
| Fula | 128 (92.09) | 11 (7.91) | | 11 (7.91) | 128 (92.09) | |
| Serer | 72 (91.14) | 7 (8.86) | | 8 (10.13) | 71 (89.87) | |
| Jola | 14 (87.50) | 2 (12.50) | | 0 (0) | 16 (100) | |
| Malinke | 16 (94.12) | 1 (5.88) | | 3 (17.65) | 14 (82.35) | |
| Soninke | 9 (100) | 0 (0) | | 3 (33.33) | 6 (66.67) | |
| Mauri | 7 (100) | 0 (0) | | 0 (0) | 7 (100) | |
| **Others †** | 14 (77.78) | 4 (22.22) | | 2 (11.12) | 16 (88.88) | |
| **Accommodation type** | | | 0.73 | | | 0.76 |
| Halls of residence | 127 (92.03) | 11 (7.97) | | 12 (8.69) | 126 (91.31) | |
| Family home | 447 (93.32) | 32 (6.68) | | 48 (10.02) | 431 (89.98) | |
| **Family members** | | | 0.54 | | | 0.32 |
| 1–2 | 36 (97.30) | 1 (2.70) | | 2 (5.40) | 35 (94.60) | |
| 3–5 | 227 (91.90) | 20 (8.10) | | 29 (11.75) | 218 (88.25) | |
| 6–8 | 176 (93.62) | 12 (6.38) | | 20 (10.64) | 168 (89.36) | |
| 9 or plus | 78 (90.70) | 8 (9.30) | | 5 (5.81) | 81 (94.19) | |
| **Matrimonial Status** | | | 0.062 | | | 0.53 |
| Single | 524 (93.9) | 34 (6.1) | | 54 (9.68) | 504 (90.32) | |
| Married | 51 (85) | 9 (15) | | 7 (11.67) | 53 (88.33) | |

*(Continued)*

**Table 2.** (Continued)

| Variable | IgM | | | IgG | | |
|---|---|---|---|---|---|---|
| | Seronegative N (%) | Seropositive N (%) | *p-value* | Seronegative N (%) | Seropositive N (%) | *p-value* |
| Divorced | 2 (100) | 0 (0) | | 0 (0) | 2 (100) | |
| **Smoker** | | | 0.08 | | | 0.16 |
| Yes | 10 (100) | 0 (0) | | 0 (0) | 14 (100) | |
| No | 503 (93.49) | 35 (6.51) | | 57 (9.69) | 531 (90.31) | |
| Stopped > 1 year | 8 (80) | 2 (20) | | 0 (0) | 10 (100) | |
| **Alcohol consumer** | | | 0.12 | | | **0.038** |
| Yes | 19 (90.48) | 2 (9.52) | | 1 (4.76) | 20 (95.24) | |
| No | 516 (93.14) | 38 (6.86) | | 51 (9.20) | 503 (90.80) | |
| Stopped > 1 year | 3 (100) | 0 (0) | | 1 (33.34) | 2 (66.66) | |
| **BMI Categorized** | | | 0.089 | | | 0.55 |
| Underweight | 67 (97.10) | 2 (2.90) | | 6 (8.70) | 63 (91.30) | |
| Normal weight | 224 (93.33) | 16 (6.67) | | 28 (11.67) | 212 (88.33) | |
| Overweight /Obesity | 46 (86.79) | 7 (11.36) | | 8 (15.09) | 45 (84.91) | |
| **Blood Group** | | | 0.95 | | | 0.094 |
| A- | 8 (88.89) | 1 (11.11) | | 1 (11.11) | 8 (88.89) | |
| A+ | 131 (93.57) | 9 (6.43) | | 14 (10) | 126 (90) | |
| AB- | 1 (100) | 0 (0) | | 1 (100) | 0 (0) | |
| AB+ | 24 (96) | 1 (4) | | 3 (12) | 22 (88) | |
| B- | 4 (80) | 1 (20) | | 0 (0) | 5 (100) | |
| B+ | 94 (92.16) | 8 (7.84) | | 7 (6.86) | 95 (93.14) | |
| O- | 22 (91.67) | 2 (8.33) | | 1 (4.17) | 23 (95.83) | |
| O+ | 279 (93) | 21 (7) | | 29 (9.67) | 271 (90.33) | |
| **COVID-19 Vaccination** | | | 0.624 | | | **<0.0001** |
| Yes | 203 (91.85) | 18 (8.15) | | 7 (3.17) | 214 (96.83) | |
| No | 340 (95.15) | 25 (6.85) | | 49 (13.42) | 316 (86.58) | |

*Statistics were calculated without missing values. Missing values were excluded from the analysis

(69.70%), declared experiencing symptoms consistent with COVID-19 in the 15 days before the survey, with common symptoms including cough (25.43%), Fatigue/Weakness (39.56%), fever (12.87%), sore throat (14.91%) and headache (50.08%). Finally, 140/637 (21.98%) participants were asymptomatic (Fig 3B).

Concerning the seroprevalence of IgM and IgG antibodies based on COVID-19 diagnosis and clinical symptoms, we found those who did not self-diagnose the disease had a higher IgM seropositivity [10.57% (95% CI: 7.88–12.59)] compared to those who did self-diagnose [5.24% (95% CI: 3.51–6.97)] (Table 3). Regarding clinical symptoms, we found no significant differences in terms of IgM seropositivity, but there was a higher IgG seroprevalence in participants reporting fatigue symptoms [92.06% (95% CI: 89.96–94.16)] compared to those who did not [80.39% (95% CI: 77.31–83.47)] with a significant p = 0.0027.

## Risk factors associated with SARS-CoV-2 IgM and IgG seropositivity among PHS community

Multiple Logistic regression analysis didn't show an association between IgG seropositivity and the factors studied. Interestingly, an association between IgM seropositivity and participants' ethnic group [O.R. 0.723 95% CI: 0.49–0.96, (*p* = 0.046)] was reported (Table 4). Using a

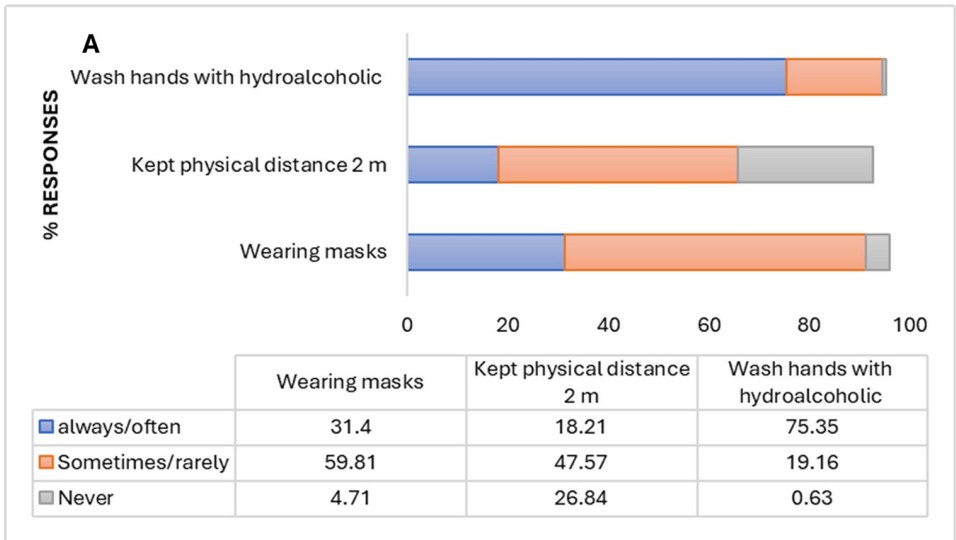

| | Wearing masks | Kept physical distance 2 m | Wash hands with hydroalcoholic |
|---|---|---|---|
| ■ always/often | 31.4 | 18.21 | 75.35 |
| ■ Sometimes/rarely | 59.81 | 47.57 | 19.16 |
| ■ Never | 4.71 | 26.84 | 0.63 |

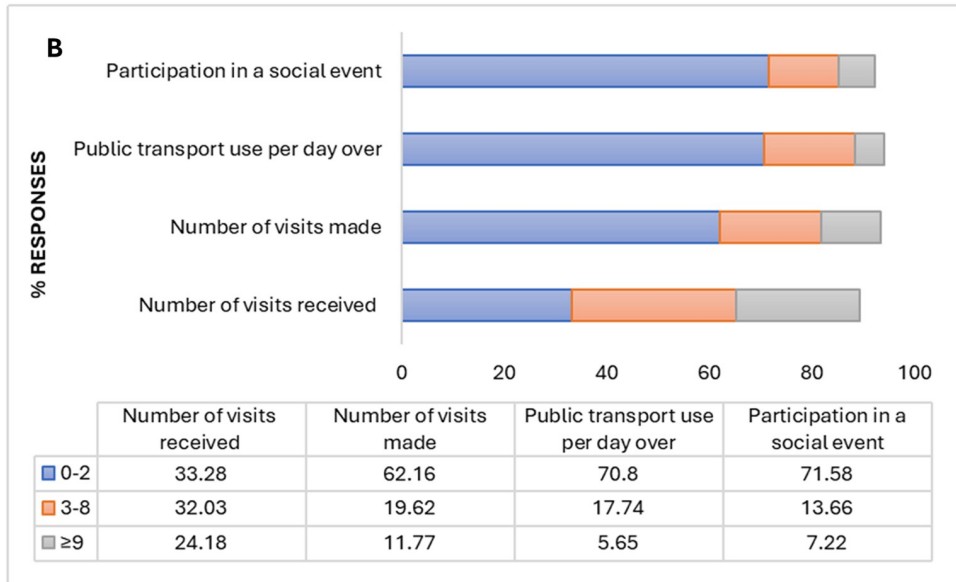

| | Number of visits received | Number of visits made | Public transport use per day over | Participation in a social event |
|---|---|---|---|---|
| ■ 0-2 | 33.28 | 62.16 | 70.8 | 71.58 |
| ■ 3-8 | 32.03 | 19.62 | 17.74 | 13.66 |
| ■ ≥9 | 24.18 | 11.77 | 5.65 | 7.22 |

**Fig 2. Compliance with preventive measures recommended by authorities.** An interviewer-administered questionnaire was distributed to participants, containing questions about compliance with WHO's recommended preventive measures, and then the rate of responses was determined. **A)** Responses related to the frequency of personal hygiene and physical distancing; and **B)** social distancing in the last 15 days before the survey. Values below bars indicate specific percentages.

univariate analysis, we found a significant association between IgM seropositivity and marital status [O.R. 2.399 95% CI: 1.08–4.87 ($p = 0.021$)], and Body Mass Indice (BMI) categorised [O.R. 2.015 95% CI: 1.12–3.56, ($p = 0.016$)]. In addition, on one hand, we found a significant association between vaccination and IgG seropositivity [O.R. 4.741 95% CI: 2.25–11.65, ($p < 0.001$)]; on the other hand, no associations were observed between SARS-CoV-2 seropositivity and sex and age groups. Regarding COVID-19 symptoms, we found a significant association between fatigue symptoms and IgG seropositivity [O.R. 2.829, 95% CI: 1.44–5.55

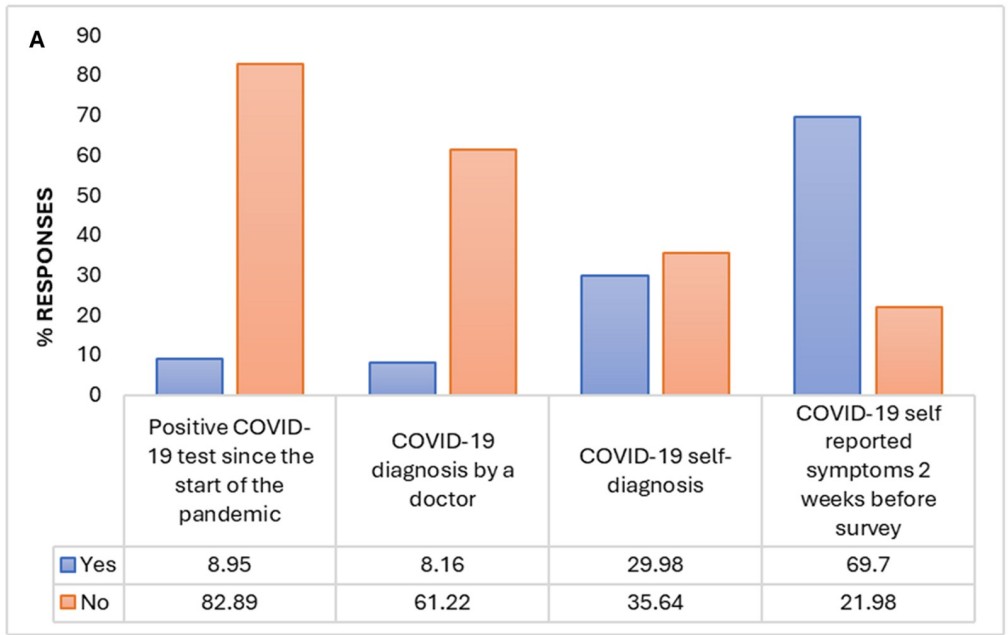

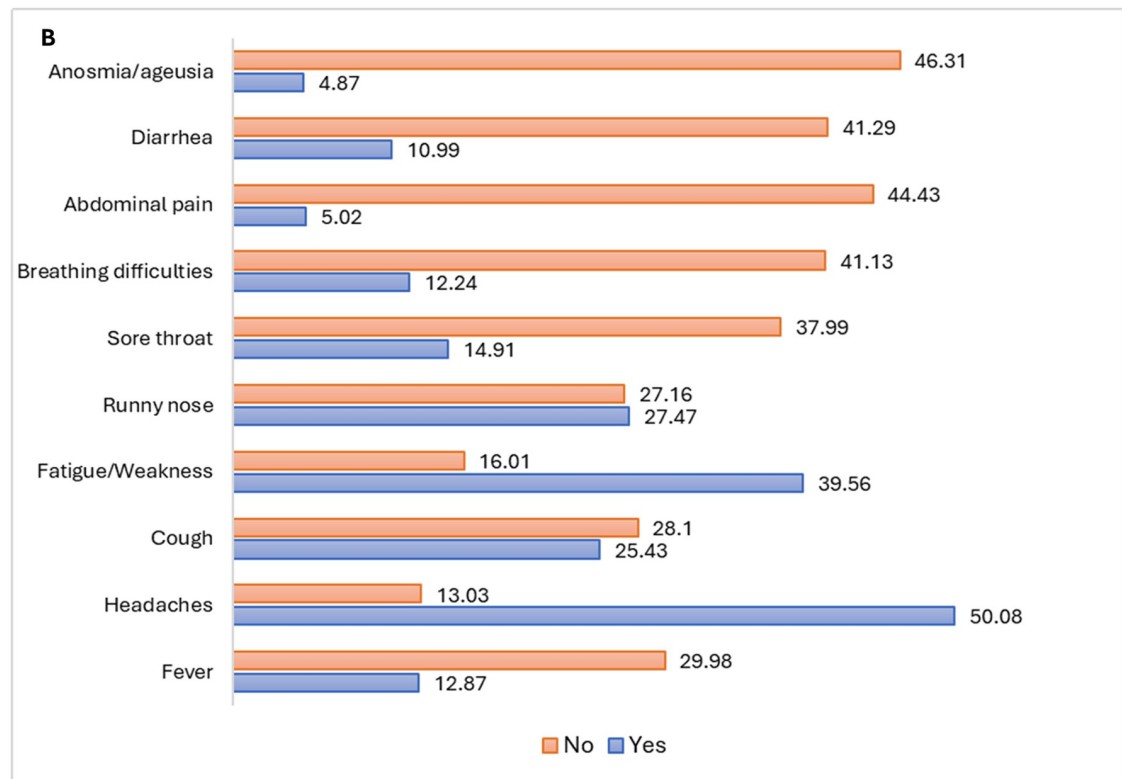

**Fig 3. Diagnosis and clinical symptoms related to COVID-19. A)** Relative frequency distribution (%) of different clinical diagnostics of COVID-19 and; **B)** symptoms of COVID-19, show that most symptoms were headaches, Fatigue or Weakness and runny nose, in respondents of our study. Notes: The values are shown in percentage.

**Table 3. Seroprevalence according to diagnosis and clinical symptoms related.**

| | IgM | | | IgG | | |
|---|---|---|---|---|---|---|
| | Seronegative N (%) | Seropositive N (%) | *p- value* | Seronegative N (%) | Seropositive N (%) | *p- value* |
| **Positive COVID-19 test since the start of the pandemic** | | | 0.411 | | | 0.814 |
| Yes | 51 (89.47) | 6 (10.53) | | 6 (10.53) | 51 (89.47) | |
| No | 493 (93.37) | 35 (6.63) | | 51 (9.66) | 477 (90.34) | |
| **COVID-19 diagnosis by a doctor** | | | 0.276 | | | 0.281 |
| Yes | 51 (98.08) | 1 (1.92) | | 8 (15.38) | 44 (84.62) | |
| No | 363 (93.08) | 27 (6.92) | | 37 (9.49) | 353 (90.51) | |
| **COVID-19 self-diagnosis** | | | **0.049 *** | | | 0.45 |
| Yes | 181 (94.76) | 10 (5.24) | | 22 (11.52) | 169 (88.48) | |
| No | 203 (89.43) | 24 (10.57) | | 20 (8.81) | 207 (91.19) | |
| **COVID-19 symptoms 2 weeks before survey** | | | 0.871 | | | 0.835 |
| Yes | 413 (93.02) | 31 (6.98) | | 46 (10.36) | 398 (89.64) | |
| No | 129 (92.14) | 11 (7.86) | | 13 (9.29) | 127 (90.71) | |
| **Fever** | | | 1 | | | 0.242 |
| Yes | 77 (93.90) | 5 (6.10) | | 6 (7.32) | 76 (92.68) | |
| No | 178 (93.19) | 13 (6.81) | | 25 (13.09) | 166 (86.91) | |
| **Headaches** | | | 0.727 | | | 0.817 |
| Yes | 298 (93.42) | 21 (6.58) | | 36 (11.28) | 283 (88.72) | |
| No | 76 (91.57) | 7 (8.43) | | 8 (9.64) | 75 (90.36) | |
| **Cough** | | | 0.804 | | | 0.509 |
| Yes | 150 (92.59) | 12 (7.41) | | 17 (10.49) | 145 (89.51) | |
| No | 168 (93.86) | 11 (6.14) | | 24 (13.41) | 155 (86.59) | |
| **Fatigue** | | | 0.372 | | | **0.0027 *** |
| Yes | 238 (94.44) | 14 (5.56) | | 20 (7.94) | 232 (92.06) | |
| No | 93 (91.18) | 9 (8.82) | | 20 (19.61) | 82 (80.39) | |
| **Runny nose** | | | 0.977 | | | 0.489 |
| Yes | 164 (93.71) | 11 (6.29) | | 19 (10.86) | 156 (89.14) | |
| No | 161 (93.06) | 12 (6.94) | | 24 (13.87) | 149 (86.13) | |
| **Sore throat** | | | 0.729 | | | 1 |
| Yes | 87 (91.58) | 8 (8.42) | | 11 (11.58) | 84 (88.42) | |
| No | 226 (93.39) | 16 (6.61) | | 28 (11.57) | 214 (88.43) | |
| Missing | 280 (93.33) | 20 (6.67) | | 22 (7.33) | 278 (92.67) | |
| **Breathing difficulties** | | | 0.61 | | | 0.23 |
| Yes | 74 (94.87) | 4 (5.13) | | 6 (7.69) | 72 (92.31) | |
| No | 242 (92.37) | 20 (7.63) | | 35 (13.36) | 227 (86.64) | |
| **Abdominal pain** | | | 0.92 | | | 0.39 |
| Yes | 30 (93.75) | 2 (6.25) | | 2 (6.25) | 30 (93.75) | |
| No | 262 (93.58) | 21 (7.42) | | 38 (13.43) | 245 (86.57) | |
| **Diarrhea** | | | 0.25 | | | 0.59 |
| Yes | 68 (97.14) | 2 (2.86) | | 7 (10) | 63 (90) | |
| No | 243 (92.39) | 20 (7.61) | | 35 (13.31) | 228 (86.69) | |
| **Anosmia/ageusia** | | | 1 | | | 0.397 |
| Yes | 29 (93.55) | 2 (6.45) | | 2 (6.45) | 29 (93.55) | |
| No | 274 (92.88) | 21 (7.12) | | 39 (13.22) | 256 (86.78) | |

**Table 4. Multiple logistic regression of sociodemographical risk factors affecting seropositivity.**

| Variable | IgM | | | | IgG | | | |
|---|---|---|---|---|---|---|---|---|
| | Univariate OR (95% CI) | p- value | Multivariate OR (95% CI) | p- value | Univariate OR (95% CI) | p-value | Multivariate OR (95% CI) | p- value |
| **Age** | 1.034 [0.99–1.06] | 0.055 | 0.971 [0,74–1,21] | 0,808 | 1.023 [0.98–1.081] | 0.345 | 0.974 [0.89–1.071] | 0.573 |
| **Gender** | 1.75 [0.943–3.25] | 0.073 | 0.776 [0,10–4.6] | 0.791 | 0.906 [0.53–1.578] | 0.728 | 0.913 [0.52–1.62] | 0.757 |
| **Occupation** | 1.652 [0.92–2.72] | 0.065 | 1.345[0,06–22.03] | 0.834 | 1.26 [0.69–2.75] | 0.508 | 2.545 [0.60–15.33] | 0.250 |
| **Level of education** | 1.526 [0.91–2.47] | 0.095 | 1.709 [0.31–9.47] | 0.532 | 0.973 [0.61–1.61] | 0.911 | 0.917 [0.48–1.73] | 0.790 |
| **Nationality** | | | 0.074 [0.003–2.21] | 0.083 | 1.96 [0.64–4.93] | 0.186 | 2.286 [0.49–7.82] | 0.225 |
| **Ethnic groups** | 0.985 [0.86–1.13] | 0.826 | 0.723 [0.49–0.96] | **0.046** * | 0.95 [0.84–1.07] | 0.426 | 0.967 [0.85–1.095] | 0.6 |
| **Place of residence** | 0.995 [0.72–1.23] | 0.740 | 0.562 [0.17–1.24] | 0.23 | 0.998 [0.80–1.27] | 0.992 | 0.981 [0.74–1.32] | 0.894 |
| **Accommodation type** | 1.21 [0.57–2.40] | 0.60 | 4.209 [0.91–21.62) | 0.067 | 1.17 [0.62–2.37] | 0.643 | 0.924 [0.41–2.23] | 0.854 |
| **Family members** | 1.133 [0.77–1.66] | 0.522 | 1.849 [0.78–4.78] | 0.175 | 1.138 [0.81–1.6] | 0.452 | 1.234 [0.83–1.85] | 0.296 |
| **Maritus status** | 2.399 [1.08–4.87] | **0.021** * | 4.44 [0.28–54.32] | 0.235 | 1.02 [0.47–2.68] | 0.956 | 1.135 [0.43–3.95] | 0.818 |
| **Smoking** | 1.004 [0.21–2.48] | 0.995 | 2.897 [0.087–36.8] | 0.457 | | | | |
| **Alcohol** | 1.064 [0.18–3.17] | 0.926 | 2.06 [0.066–30.84} | 0.622 | 0.835 [0.32–3.25] | 0.747 | | |
| **BMI Categorized** | 2.015 [1.12–3.56] | **0.016** * | 1.215 [0.39–3.73] | 0.729 | 0.785 [0.49–1.29] | 0.329 | 0.718 [0.43–1.21] | 0.206 |
| **Blood Group** | 1.011 [0.89–1.152] | 0.861 | 1.049 [0.786–1.47] | 0.235 | 1.023 [0.91–1.14] | 0.679 | 0.988 [0.86–1.24] | 0.842 |
| **Vaccination Status** | 1.206 [0.63–2.25] | 0.560 | 0.428 [0.08–1.89] | 0.285 | 4.741 [2.25–11.65] | <**0.001** * | 2.714 [0.59–48.48] | 0.325 |

OR odds ratio. Signifcant values are in bold.

(p = 0.0023)] (Table 5). Finally, no significant association were found between IgM and IgG seropositivity and preventive measures (S2 Table).

## Discussion

This study represents the first sero-epidemiological survey of COVID-19 carried out in an educational establishment in Senegal. The survey was carried out on samples collected from

**Table 5. Multiple logistic regression of diagnosis and clinical symptoms affecting seropositivity.**

| Variable | IgM | | IgG | |
|---|---|---|---|---|
| | Univariate OR (95% CI) | p- value | Univariate OR (95% CI) | p- value |
| **Positive COVID-19 test since the start of the pandemic** | 1.657 [0.60–3.87] | 0.278 | 0.909 [0.398–2.45] | 0.834 |
| **COVID-19 diagosis by a doctor** | 0.264 [0.014–1.28] | 0.95 | 0.576 [0.263–1.40] | 0.191 |
| **COVID-19 self-diagosis** | 0.467 [0.21–0.97] | 0.051 | 0.742 [0.39–1.41] | 0.360 |
| **COVID-19 symptoms 2 weeks before survey** | 0.880 [0.44–1.88] | 0.727 | 0.885 [0.45–1.65] | 0.713 |
| Fever | 0.889 [0.28–2.45] | 0.829 | 1.908 [0.80–5.306] | 0.174 |
| Cough | 1.222 [0.52–2.89] | 0.643 | 1.321 [0.68–2.59] | 0.409 |
| Fatigue | 0.608 [0.26–1.50] | 0.262 | 2.829 [1.44–5.55] | **0.0023** * |
| Headaches | 0.765 [0.33–2] | 0.556 | 0.838 [0.35–1.79] | 0.669 |
| Sore throat | 1.299 [0.51–3.06] | 0.5 | 0.999 [0.488–2.18] | 0.998 |
| Runny nose | 0.889 [0.38–2.11] | 0.807 | 1.323 [0.697–2.54] | 0.394 |
| Breathing difficulties | 0.654 [0.186–1.79] | 0.451 | 1.850 [0.80–5.046] | 0.183 |
| Abdominal pain | 0.831 [0.13–3.03] | 0.809 | 2.327 [0.66–14.74] | 0.261 |
| Diarrhea | 0.357 [0.056–1.27] | 0.172 | 1.382 [0.62–3.52] | 0.460 |
| Anosmia/ageusia | 0.899 [0.14–3.28] | 0.890 | 2.209 [0.63–14.00] | 0.291 |

personnel and students from May 2022 to August 2022, after the fourth wave of COVID-19 in Senegal. A total of 637 participants were included, with 62% women and 37% men. The median age was 21, justified by the study being conducted in a university environment. Consequently, the study population mainly consisted of students representing approximately 86% of the participants. 35% of the participants have either received partial or full vaccination, which is higher than the national vaccination rate of 16.9% at the time of the survey [33]. This could be attributed to the high level of education and awareness about the importance of vaccination among our population. Additionally, the CAD University authorities tried to set up a vaccination campaign at various sites during October 2021. However, the vaccination rate remains low compared to global levels, with nearly 70% of the population having received at least one vaccine dose since the pandemic's beginning [34].

The study showed IgG and IgM SARS-CoV-2 antibody seroprevalences of 92% (95% CI: 89.90–94.11) and 6.91% (95% CI: 4.93–8.87), respectively. Interestingly, 6.90% of participants tested negative for both IgM and IgG corresponding to persons who probably lacked antibodies following infection. Indeed, Absence of antibodies varied independently by illness severity, race/ethnicity, obesity, and immunosuppressive drug therapy. The proportion of seronegative remained relatively stable among persons tested up to 90 days post-symptom onset [35]. The IgM seroprevalence could indicate that the virus was circulating within PHS at the time of the survey. Indeed, in seroprevalence surveys, it is recommended to use IgM and IgG detection to track the spread of infection and defining herd immunity barrier and individual immunization levels in the ongoing COVID-19 pandemic [36]. IgM antibody levels rise around a week after the initial infection [22]. IgG antibodies appear later than IgM antibodies (generally within 14 days of infection). They can persist for up to a year [37, 38], meaning that IgG antibodies serve as an indicator of infection or vaccination status [39]. The high IgG seroprevalence in our study may be due to increased SARS-CoV-2 exposure and the COVID-19 vaccination policy. Our study took place after the fourth wave of COVID-19 in Senegal in January 2022, and the vaccine rollout began in February 2021 [40], which could explain our result. The first SARS-CoV-2 Senegalese seroepidemiological surveys in 2020 showed seroprevalence around 20–30% [41, 42]. Since then, studies have indicated a sharp increase in seroprevalence, possibly due to the circulation of various variants, including the deadly delta variant [11, 40, 43, 44]. Similar high seroprevalence rates have been observed in other African countries during the same period, highlighting the importance of conducting regular epidemiological surveys to track the spread of SARS-CoV-2 [45–47].

Our results also showed a different distribution by age group, with lower IgM seroprevalence in younger individuals, particularly those aged 18–25 years, compared to those aged 55–65 years (5.5% and 33.34%, respectively) with a $p < 0,001$. The 18–25 age group appears less affected by recent infections. This reinforces the data showing that young people are less affected by SARS-CoV-2 infection [48]. In Senegal, it was reported during the first wave that the seroprevalence of SARS-CoV-2 was higher in patients older than 65 years [49]. Afterwards, the study investigated IgM seroprevalence based on the type of residence. Surprisingly, there was no significant difference in IgM seroprevalence between those living in halls of residence (7.97%, 95% CI: 5.87–10.07) and those living in family homes (6.68%, 95% CI: 4.74–8.62), with a $p = 0.73$. Our study disproves the hypothesis that the environment at CAD University would create a climate conducive to spreading the virus. The PHS community, primarily made up of young people, does not appear to be contributing to new infections [48, 50].

During the survey, a small percentage of participants (8.95%) reported testing positive for COVID-19 infection through RT-PCR since the beginning of the pandemic. 8.16% were diagnosed as COVID-19 positive, and 29.98% reported having been self-diagnosed as COVID-19 positive. These results, like those of vaccination rate, contrast with the high seroprevalence of

IgG, which was over 90% for the study population. In Senegal, limited access to rapid RT-PCR diagnostic tests has hindered COVID-19 detection [51]. WHO reported that only one in seven COVID-19 infections in Africa are being detected, emphasizing the need to expand testing capabilities [52]. Senegal, with a population of 17 million, has conducted about 1.5 million tests, whereas the USA, with a population of around 335 million, has administered over 880 million tests [53].

Approximately 70% of the participants showed symptoms associated with COVID-19 at the time of the survey. The most common symptoms reported were cough, fatigue/weakness, fever, sore throat, and headache. According to China CDC data, individuals with COVID-19 have experienced a wide range of symptoms, varying from mild to severe illness. Symptoms may appear 2–14 days after exposure to the virus. However, these symptoms can be similar to those of the flu caused by influenza [54]. In Senegal, influenza circulates year-round with two peaks: January to March and August to October. However, the expected January–March peak in 2021–2022 disappeared due to active SARS-CoV-2 circulation. An unexpected influenza peak was observed from May to July 2022, suggesting potential viral interference that needs further investigation in tropical settings [55].

In our study, we observed a low level of compliance with the preventive measures recommended by health authorities such as wearing masks (31.40%) and keeping physical distance (17.42%). However, most participants reported a high level of compliance with hand washing (75.35%). Our study found opposing results compared to Kearney et al., who noted significant adherence to COVID-19 prevention measures among Senegalese participants, especially regarding wearing masks and practising good personal hygiene [56]. The variance in our findings may be attributed to the fact that the individuals in our research are younger and less inclined to adhere to preventive protocols [57, 58]. Our findings showed that IgM seropositivity depends on the number of visits to others, indicating a lack of social distancing (p = 0.008). This highlights the importance of following the World Health Organization's guidelines on extensive social distancing as a non-pharmaceutical way to reduce the spread of infection and related deaths [59]. However, research has indicated that the effectiveness of social distancing measures varies based on demographic, environmental, and economic factors [60].

We investigated the association of sociodemographic, clinical and lifestyle factors, such as BMI, blood type, smoking and alcohol consumption, with seropositivity using logistics. Then, we found an association between BMI categorised and IgM seropositivity (O.R. 0.238, $p = 0.043$). Previous studies have shown that obese individuals (BMI $> 30$ kg/m$^2$) were significantly more likely to be seropositive. It is uncertain whether the raised seroprevalence in these groups represents a greater risk of SARS-CoV-2 infection. However, obese individuals are known to experience more severe COVID-19 symptoms [61, 62]. Interestingly, a significant association between IgM seropositivity and ethnic group was found (O.R. 0.723, $p = 0.046$), suggesting a difference in susceptibility to SARS-CoV2- depending on race and ethnicity. According to previous findings, it seems there are disparities in the effects of COVID-19 infection among different racial/ethnic groups [63, 64]. Then, understanding the mechanisms for the disparity will help to evaluate the risk for COVID-19 according to ethnicity.

Our findings showed that IgM seropositivity was associated with marital status (unmarried/married) (O.R. 2.39, $p = 0.021$); another study reveals that transmission of SARS-CoV-2 is variable among different people within the home. For instance, the risk for infection was higher between spouses, at 43%, which could be a reflection of transmission through intimacy or longer or more direct exposure [65]. Regarding vaccination status, we found a relationship between vaccination and IgG seroprevalence (O.R. 4.741, $p < 0.001$), showing that people who had received vaccine doses were four times more likely to produce anti-SARS-CoV-2 antibodies than those who had not. It will be interesting to determine the IgG-neutralizing antibodies

and to determine antibody levels according to vaccination status and type. Indeed, it has been shown that vaccines from different manufacturers might induce different antibody responses [66].

Some limitations of our study must be considered. By design, we carried out an on-site university population-based cross-sectional study. Thus, the results cannot be extrapolated directly to the general CAD University population. Participants were subjected to recall bias when completing the questionnaires, particularly preventive measures at the time of contact. We also noted a very low level of participation from professors and TAS officers. This constitutes a bias in the statistical calculations. Missing data for certain criteria and non-prefer responders were a limitation of statistics. To date, our study is the only SARS-CoV-2 seroepidemiological survey conducted in a university community in Senegal. Data on the Serological testing performed by measuring IgM and IgG immunoglobulins against the SARS-CoV-2 targeting peak S1 protein separately is a strength of our study. This allowed us to differentiate recent from previous infections. Finally, with our questionnaire, we were able to collect a lot of data, which was useful in calculating associations with risk factors.

## Conclusions

In summary, our analysis of more than 630 subjects from Polytechnic High School community members estimated the extent of exposure to SARS-CoV-2. Our study revealed a high seroprevalence from May to August 2022 following the fourth wave of COVID-19 in Senegal. Our results show that the majority of students and staff have already been exposed to SARS-CoV-2 and confirm the circulation of the virus (SARS-CoV-2) at the time of the survey as shown by the high IgM and IgG seroprevalence. The data show a link between seropositivity and various factors such as age and non-compliance with prevention measures and Diagnosis and Clinical symptoms. These results underline the importance of seroepidemiological surveys to estimate the real impact of the COVID-19 pandemic and the disparities between populations to establish a profile of the transmission dynamics of the virus. In addition, these results may be essential for the CAD university in the event of the emergence of future waves to make appropriate decisions and put in place means of monitoring the evolution of the pandemic after the relaxation of social distancing measures and the implementation of a vaccination schedule at CAD University, which will serve as a basis for other universities in Senegal.

## Supporting information

**S1 Fig. Distribution of different antibody response profiles in the study population.** The vertical axis (y-axis) represents the IgM/IgG SARS-CoV-2 antibody rate. The horizontal axis (x-axis) represents the different profiles of IgM/IgG responses.
(PDF)

**S1 Table. Distribution of Sars-Cov-2 Ig M and G seropositive and seronegative according to preventive measures.**
(DOCX)

**S2 Table. Multiple logistic regression of preventive measures affecting seropositivity.**
(DOCX)

## Acknowledgments

We thank the top management of the "Ecole Supérieure Polytechnique de Dakar" and all students, Technicians, Administrative, Service officers and professors members who voluntarily participated in our study.

## Author Contributions

**Conceptualization:** Fatou THIAM, Abou Abdallah Malick DIOUARA, Idy DIOP, Saidou Moustapha SALL, Massamba DIOUF, Cheikh Momar NGUER.

**Data curation:** Fatou THIAM, Clemence Stephanie Chloe Anoumba NDIAYE, Khadim KEBE, Djibaba DJOUMOI, Sarbanding SANE, Seynabou COUNDOUL, Sophie Deli TENE, Abdou Lahat DIENG, Mamadou NDIAYE.

**Formal analysis:** Fatou THIAM, Clemence Stephanie Chloe Anoumba NDIAYE, Ibrahima DIOUF, Khadim KEBE, Assane SENGHOR, Djibaba DJOUMOI, Mame Ndew MBAYE, Idy DIOP, Sarbanding SANE, Seynabou COUNDOUL, Sophie Deli TENE, Abdou Lahat DIENG, Mamadou NDIAYE.

**Funding acquisition:** Fatou THIAM, Abou Abdallah Malick DIOUARA, Ibrahima DIOUF, Khadim KEBE, Idy DIOP, Saidou Moustapha SALL, Massamba DIOUF, Cheikh Momar NGUER.

**Investigation:** Fatou THIAM, Assane SENGHOR, Djibaba DJOUMOI, Mame Ndew MBAYE, Sarbanding SANE, Seynabou COUNDOUL, Sophie Deli TENE, Mamadou DIOP, Massamba DIOUF, Cheikh Momar NGUER.

**Methodology:** Fatou THIAM, Abou Abdallah Malick DIOUARA, Clemence Stephanie Chloe Anoumba NDIAYE, Ibrahima DIOUF, Assane SENGHOR, Djibaba DJOUMOI, Sarbanding SANE, Seynabou COUNDOUL, Sophie Deli TENE, Mamadou DIOP, Massamba DIOUF.

**Project administration:** Fatou THIAM.

**Resources:** Fatou THIAM.

**Software:** Clemence Stephanie Chloe Anoumba NDIAYE, Ibrahima DIOUF, Mame Ndew MBAYE, Abdou Lahat DIENG.

**Supervision:** Fatou THIAM, Abou Abdallah Malick DIOUARA, Mamadou DIOP, Saidou Moustapha SALL, Cheikh Momar NGUER.

**Validation:** Fatou THIAM, Abou Abdallah Malick DIOUARA, Mamadou DIOP, Cheikh Momar NGUER.

**Visualization:** Fatou THIAM, Clemence Stephanie Chloe Anoumba NDIAYE, Ibrahima DIOUF, Mamadou DIOP, Abdou Lahat DIENG.

**Writing – original draft:** Fatou THIAM.

**Writing – review & editing:** Fatou THIAM, Abou Abdallah Malick DIOUARA, Ibrahima DIOUF, Khadim KEBE, Mame Ndew MBAYE, Idy DIOP, Mamadou DIOP, Abdou Lahat DIENG, Saidou Moustapha SALL, Massamba DIOUF, Cheikh Momar NGUER.

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
