## [Decision Letter · Decision Letter 0]

24 Jul 2024

PONE-D-23-44123Serological Survey in a university community after the fourth wave of COVID-19 in SenegalPLOS ONE

Dear Dr. thiam,

Thank you for submitting your manuscript to PLOS ONE. After careful consideration, we feel that it has merit but does not fully meet PLOS ONE’s publication criteria as it currently stands. Therefore, we invite you to submit a revised version of the manuscript that addresses the points raised during the review process.

We look forward to receiving your revised manuscript.

Kind regards,

José Ramos-Castañeda, M.Sc., Ph.D

Academic Editor

PLOS ONE

“This research was funded by Polytechnique High School competitive research impulse fund.”

“The authors declare no conflict of interest. The funders had no role in the design of the study; in the collection, analyses, or interpretation of data; in the writing of the manuscript; or in the decision to publish the results”.”

5. Please provide a complete Data Availability Statement in the submission form, ensuring you include all necessary access information or a reason for why you are unable to make your data freely accessible. If your research concerns only data provided within your submission, please write "All data are in the manuscript and/or supporting information files" as your Data Availability Statement.

7. Please include a separate caption for each figure in your manuscript.

8. Please include your tables as part of your main manuscript and remove the individual files. Please note that supplementary tables (should remain/ be uploaded) as separate "supporting information" files

Reviewers' comments:

Reviewer's Responses to Questions

**Comments to the Author**

1. Is the manuscript technically sound, and do the data support the conclusions?

Reviewer #1: Yes

Reviewer #2: Partly

2. Has the statistical analysis been performed appropriately and rigorously? 

Reviewer #1: Yes

Reviewer #2: I Don't Know

3. Have the authors made all data underlying the findings in their manuscript fully available?

Reviewer #1: Yes

Reviewer #2: Yes

4. Is the manuscript presented in an intelligible fashion and written in standard English?

Reviewer #1: Yes

Reviewer #2: Yes

5. Review Comments to the Author

Reviewer #1: Experiments, statistics, and other analyses are performed to a high technical standard and are described in sufficient detail.

Conclusions are presented in an appropriate fashion. however, the study's generalizability and dependability have shortcomings on both a national and worldwide scale.

Therefore, the paper needs minor revision before publication and make it sounder.

Reviewer #2: Thiam and Cols conducted a study to evaluate the seroprevalence of IgM and IgG antibodies against SARS-CoV-2 among students and staff of the Polytechnic High School (PHS) in Senegal. They also conducted a questionnaire to evaluate the associations between seropositivity and different parameters.

The study is interesting because it provides new information from Senegal. Still, it is also limited because it only evaluates the IgM and IgG, which can be compensated with the information from the questionnaire. However, the authors overestimate the results and make several conclusions that lack support. The discussion is too long!! Please reduce it to two pages (50% reduction).

Lines 207-211. Please correct the statement because it is not true. IgM was a marker of a recent infection but only in the early phase of the pandemic. IgG is not a marker of old infection in a vaccinated population.

Line 263. Why did the authors not include the question "previous infection" or "symptoms of COVID-19."

Lines 278-279. The IgM positive were also IgG positive? Do you found IgM and IgG negative?

Line 291. Please double-check lines 291 to 306; I can't find congruency with the results in the table. For example, in line 292, it is unclear what 95.24% and 90.80% mean. I can't find these results in the table. Similar for the following lines.

Line 327.328. Why did the authors not ask if the participants had previous infections or symptoms suggesting COVID-19? About 30% were vaccinated, so this statement can't be supported with these data.

Line 355-339. Remove it, this statement is incorrect.

Line 341-342. "or vaccination"

Line 370-372. Remove it, this statement is incorrect.

I suggest remove lines 370-379

Lines 388-389. At least confirming that those patiens were also negative for IgG

Lines 395-397- PCR studies must be done to support this hypothesis. otherwise remove

The rest of the discussion need an improvement.

6. PLOS authors have the option to publish the peer review history of their article (what does this mean?). If published, this will include your full peer review and any attached files.

Reviewer #1: **Yes: **Silamlak Birhanu Abegaz

Reviewer #2: **Yes: **Jesús Hernández

---

## [Author Response · Author response to Decision Letter 0]

26 Sep 2024

Dear reviewers,

We thank you for your generous comments on our manuscript and have edited it to address their concerns.

as you have recommended, we have substantially improved the text and corrected the English and the phrasing of specific sentences to make them easier to understand. You'll find answers to your comments, suggestions and questions above.

We believe that our manuscript revised is now suitable for publication in Plos One.

Reviewer 1

Experiments, statistics, and other analyses are performed to a high technical standard and are described in sufficient detail.

Conclusions are presented in an appropriate fashion. however, the study's generalizability and dependability have shortcomings on both a national and worldwide scale.

Therefore, the paper needs minor revision before publication and make it sounder.

Dear reviewer, thank you for reading and appreciating our work. 

Thank you for this remark. We have substantially improved the manuscript to make it sounder

Reviewer 2

Thiam and Cols conducted a study to evaluate the seroprevalence of IgM and IgG antibodies against SARS-CoV-2 among students and staff of the Polytechnic High School (PHS) in Senegal. They also conducted a questionnaire to evaluate the associations between seropositivity and different parameters.

The study is interesting because it provides new information from Senegal. Still, it is also limited because it only evaluates the IgM and IgG, which can be compensated with the information from the questionnaire. However, the authors overestimate the results and make several conclusions that lack support. The discussion is too long!! Please reduce it to two pages (50% reduction).

Dear reviewer, thank you for this point. As we you recommended we revised the discussion to improve and make it easier to read. We also include “previous infection and symptoms of COVID-19

Lines 207-211. Please correct the statement because it is not true. IgM was a marker of a recent infection but only in the early phase of the pandemic. IgG is not a marker of old infection in a vaccinated population.

Thank you, this statement was corrected (Line 191 – 196).

Line 263. Why did the authors not include the question "previous infection" or "symptoms of COVID-19."

Dear reviewer, thank you for reading and questioning. We had initially included this data, but then chose to remove it at the suggestion of some of the co-authors. We thank you for raising this point, as the data is important for a better understanding of our results. We have therefore added figure 2 and tables 4and 6 relating to “previous infection” and 'symptoms Covid”. And as mentioned above, we've added this part to the discussion

Lines 278-279. The IgM positive were also IgG positive? Do you found IgM and IgG negative?

Dear reviewer, thank you for this, we add any information concerning this point. In results section , we added the data concerning these points(Line 246 – 248). These results were discussed.

Line 291. Please double-check lines 291 to 306; I can't find congruency with the results in the table. For example, in line 292, it is unclear what 95.24% and 90.80% mean. I can't find these results in the table. Similar for the following lines.

Here we analyzed the differences in IgG seroprevalence between alcohol users, non-users and those who had stopped drinking for more than a year. We've revised the results section to make it easier to understand (line 262 – 265).

Line 327.328. Why did the authors not ask if the participants had previous infections or symptoms suggesting COVID-19? About 30% were vaccinated, so this statement can't be supported with these data.

the question previous infection and symptoms has been taken into account in the questionnaires. the data is available. We had chosen not to include them in the manuscript and to publish them separately, but these data have been added to support our results.

Line 355-339. Remove it, this statement is incorrect.

This statement is revised et corrected (Line 336 – 345).

Line 341-342. "or vaccination"

Thank you for this suggestion. It is added (Line 347 – 348)

Line 370-372. Remove it, this statement is incorrect.

I suggest remove lines 370-379

As you suggested, we remove this part

Lines 388-389. At least confirming that those patients were also negative for IgG

Lines 395-397- PCR studies must be done to support this hypothesis. otherwise remove

The rest of the discussion need an improvement.

The entire discussion has been reviewed to remove any ambiguities and clarify our ideas and conclusions.

---

## [Decision Letter · Decision Letter 1]

23 Oct 2024

Serological Survey in a university community after the fourth wave of COVID-19 in Senegal

PONE-D-23-44123R1

Dear Dr. thiam,

We’re pleased to inform you that your manuscript has been judged scientifically suitable for publication and will be formally accepted for publication once it meets all outstanding technical requirements.

Kind regards,

José Ramos-Castañeda, M.Sc., Ph.D

Academic Editor

PLOS ONE

Additional Editor Comments (optional):

Reviewers' comments:

Reviewer's Responses to Questions

**Comments to the Author**

1. If the authors have adequately addressed your comments raised in a previous round of review and you feel that this manuscript is now acceptable for publication, you may indicate that here to bypass the “Comments to the Author” section, enter your conflict of interest statement in the “Confidential to Editor” section, and submit your "Accept" recommendation.

Reviewer #2: (No Response)

2. Is the manuscript technically sound, and do the data support the conclusions?

Reviewer #2: Yes

3. Has the statistical analysis been performed appropriately and rigorously? 

Reviewer #2: I Don't Know

4. Have the authors made all data underlying the findings in their manuscript fully available?

Reviewer #2: Yes

5. Is the manuscript presented in an intelligible fashion and written in standard English?

Reviewer #2: No

6. Review Comments to the Author

Reviewer #2: I have some additional comments:

In Table 1, the percentages of BMI are incorrect; please change accordingly.

In lines 259-263 and 387-389, the percentages of IgM+, IgG+, and IgM-IgG- are wrong. Please check and correct them accordingly.

Line 319 ... During the survey, a low rate of respondents (8.95%) declared...

The authors should spell out numbers at the beginning of a sentence rather than using a numera.

7. PLOS authors have the option to publish the peer review history of their article (what does this mean?). If published, this will include your full peer review and any attached files.

Reviewer #2: **Yes: **Jesús Hernández

---

## [Editor Report · Acceptance letter]

5 Nov 2024

PONE-D-23-44123R1 

PLOS ONE

Dear Dr. thiam, 

I'm pleased to inform you that your manuscript has been deemed suitable for publication in PLOS ONE. Congratulations! Your manuscript is now being handed over to our production team.

Kind regards, 

on behalf of

Dr. José Ramos-Castañeda 

Academic Editor

PLOS ONE